# Information System Success for Organizational Sustainability: Exploring the Public Institutions in Saudi Arabia

**Abdullah Almuqrin [1,\*], Ibrahim Mutambik [1], Abdulaziz Alomran [1] and Justin Zuopeng Zhang [2]**

[1] Department of Information Science, College of Humanities and Social Sciences, King Saud University, Riyadh 11451, Saudi Arabia; imutambik@ksu.edu.sa (I.M.); benomran@ksu.edu.sa (A.A.)

[2] Department of Management, Coggin College of Business, University of North Florida, 1 UNF Drive, Building 42, Jacksonville, FL 32224, USA; justin.zhang@unf.edu

\* Correspondence: aalmogren@ksu.edu.sa

**Abstract:** Organizational sustainability supports the financial, social, and cultural well-being of organizations and their surrounding communities. However, few studies have examined organizational sustainability in Saudi Arabia or its link to information technology. This study used self-reported data from a large sample of employees at various Saudi government institutions to conclude that these institutions moderately implemented organizational sustainability. Correlation and regression analyses demonstrated weak associations between various types of organizational sustainability and dimensions of information system success, where user satisfaction with information systems is the strongest positive predictor of perceived organizational sustainability. Organizational sustainability is still emerging in the public sector, and further research is needed to identify predictors of its success.

**Keywords:** organizational sustainability; information system success; Saudi Arabia; public sector; regression analysis

## 1. Introduction

Organizational sustainability has received considerable attention among scholars and practitioners [1–3]. Most studies have defined it in reference to the triple-bottom-line conceptualization as the preservation of social, economic, and environmental resources for future generations at the organizational level [4]. More specifically, the social dimension of organizational sustainability could be described as respecting, incentivizing, fairly paying, providing benefits to, and educating workers in organizations [2]. Furthermore, companies should consider ethics, intergenerational justice, and systems for gratification to ensure social sustainability [1]. With regard to the economic dimension, organizational sustainability could be defined as an organization's ongoing cash flow, long-term financial stability, stakeholder profit security, job offer consistency, market expansion/repositioning, and competitiveness [5]. Finally, environmental sustainability includes biodiversity preservation, ensuring the ability of natural resources to regenerate, and waste/emissions reduction [2,4].

Organizational sustainability thus offers a plethora of benefits to a business socially, economically, and environmentally. According to one study, increasing sustainability exploitation increased financial and market performance by 43.2% among European companies [6]. Similarly, organizations with social responsibility initiatives have scored four times higher than those without concrete societal links in terms of stakeholder and client satisfaction [7]. Additionally, companies disclosing sustainability measures have shown a 55% increase in employee morale, a 43% increase in business process efficiency, and a 38% increase in employee loyalty [8]. In another study, companies implementing subsidies and tax break programs to ensure environmental sustainability had better success rates [9]. Furthermore, taking measures to decrease greenhouse gas emissions could ensure the sustainability of companies for future generations [10].

Despite the benefits outlined above, many organizations lack sustainable practices in these three dimensions. In a survey of S&P 500 companies, only 60% reported having an organizational sustainability policy in place [11]. In addition, the Governance & Accountability Institute found that only 49% of Russell 1000 companies publicly reported their sustainability metrics [12]. Clinton and Whisnant recommended more organizational sustainability in streaming services to counter their extravagant financial waste and carbon emissions [13]. In the fast-food industry, employee and stakeholder satisfaction has been shown to be scarce, implying the need for more economic sustainability measures [14]. In the same vein, little attention has been given to multiple dimensions of sustainability in tech companies [15].

Scholars have argued that adopting new technology, such as interactive intelligent cloud-based information systems, makes it easier to implement cost-effective sustainability measures across an organization [8,9,16,17]. However, there is little empirical research on the association between technology and organizational sustainability. This lack of studies is even more pronounced in emerging and developing economies, such as Saudi Arabia.

Organizational sustainability research has been limited by the difficulty of setting parameters for case studies in Saudi Arabia [18], with a lack of periodic assessments following organizational sustainability studies, limiting long-term research [19]. Additionally, authors have been discouraged from writing about sustainability by businesses, the public, and the government in Saudi Arabia due to a social desire to mask its areas of scarcity [20]. Instead, short-term economic studies have taken precedence over sustainability [21]. Furthermore, many business executives in Saudi Arabia are foreigners disinterested in assisting with organizational sustainability research in the country [22]. The examination of organizational sustainability in Saudi Arabia is further limited because many businesses do not report their annual statistics as usable data for organizational sustainability measurements [23].

In order to address this gap in the literature, the current study examined the extent to which Saudi public institutions exhibited organizational sustainability practices. A large convenience sample of Saudi public employees responded to a validated online questionnaire, which elicited their perceptions of their institutions' implementation of social, economic, environmental, cultural, and administrative sustainability. In addition, the study established cross-sectional associations between various measures of technology implementation, information system success dimensions, and organizational sustainability practices. Multiple regression analysis was carried out to investigate the extent to which perceived information system success affected perceptions of organizational sustainability among Saudi public organizations.

The next section examines prior conceptualizations of organizational sustainability, information system success, and the measurement of these concepts. After that, the methodology explains the sampling, data collection, instruments, and data analysis. The main findings are then presented using descriptive statistics, and the empirical results are presented based on multiple regression analysis. The Section 5 discusses the study's relevance to past research, theoretical implications, practical recommendations, future research directions, and limitations of the study.

*Research Questions and Objectives*

1. To what extent do Saudi Arabian institutions practice organizational sustainability?
2. To what extent does technology implementation influence organizational sustainability among Saudi Arabian institutions?

This paper aims to review the extent to which Saudi Arabian institutions implement organizational sustainability practices. Furthermore, this investigation attempts to capture the association linking technology implementation to organizational sustainability. More specifically, information systems' success and predictive power on organizational sustainability implementation are examined from more than a single prism.

## 2. Literature Review

### 2.1. Conceptualization of Organizational Sustainability

Given the varying definitions for organizational sustainability, conceptualizations of the construct have covered many dimensions. Hansmann et al. divided organizational sustainability into three pillars: economic, environmental, and social sustainability [24]. These dimensions refer to the triple-bottom-line perspective [4,25,26]. For a company to adopt significant economic sustainability, plans for economic success and viability must be established [27,28]. Additionally, the implementation of environmental sustainability at a company must follow relevant government regulations [29,30]. Similarly, Munck et al. and Callado considered an organization compliant with environmental organizational sustainability if it avoided having a negative impact on the environment [28,31]. Finally, social organizational sustainability compliance encompasses a range of managerial actions to improve employee and business partner loyalty [1,27–30].

Other researchers have divided organizational sustainability into different dimensions, such as Savytska et al., who outlined six levels [17]. Level 0 includes organizations with no regard for sustainability, Level 1 indicates compliance with sustainability legislation, Level 2 includes businesses that try to be sustainable for the sake of their reputation, and Level 3 recognizes an organization as going beyond legal requirements to encompass equality within the triple-bottom-line. Level 4 consists of organizations that view sustainability as inevitable and use creative techniques to prepare for the triple-bottom-line. Finally, Level 5 entails making decisions based on the interdependence between people, the organization, the environment, and the economy.

Another organizational sustainability classification system utilized by the International Sustainability Council proposed two dimensions by evaluating internal and external factors [32–34]. This organizational sustainability model (OSM) divides internal factors into labor, investment, resource use, procurement, business model, and culture. The external factors are environment, society, and ethics. Organizations may only be certified as "sustainable" if they meet a minimum threshold after their values regarding these categories are evaluated [33].

One last organizational sustainability measurement encompasses a seven-dimensional social sciences approach [16,17]. The first four dimensions consist of financial sustainability regarding profitability and employee satisfaction, social attention to relationships with customers and stakeholders, the implementation of organizational best measures and growth systems for company members, and transformative processes encouraging personal and business-related innovation [15,35]. Wright and Nyberg explained the importance of integrity and culture within the fifth dimension of this framework, followed by community involvement and environmental awareness [36]. The final dimension involves the values of a company, where researchers and fieldworkers concur that social justice, global initiatives, and human rights are of utmost importance [16,34].

### 2.2. Measurement of Organizational Sustainability

Various methods have been proposed to measure organizational sustainability, but few studies have used questionnaires to operationalize it. For example, Balasubramanian and Balaji measured organizational sustainability using a 26-item questionnaire completed by a snowball sample of employees [37]. They analyzed organizational sustainability as a six-dimensional construct consisting of environmental management, employee-related, financial, public sustainability, emission control measures, and governance. Similarly, Santos et al. used Likert-scale items to measure organizational sustainability, finding a five-dimensional structure consisting of posture, direction, organization, behavior, and evaluation [38]. Yu et al. proposed four factors to measure organizational sustainability on a Likert scale [39]. These included eight employee-organization relationship items, seven organizational trust items, eight innovative climate items, and six innovative behavior items.

Other researchers have measured organizational sustainability using indices of best practices or essential components. Nawaz and Koç, for instance, identified nine func-

tional areas with varying best practices to investigate whether organizations possessed sustainability [40]. These areas were optimizing resources and minimizing emissions, operational excellence, social development and corporate culture, innovation and development, logistics and procurement, governance, sustainability management tools, relationships between employees, and the overall health, safety, and security of the company. Similarly, Mills et al. measured organizational sustainability using an index of eight components: ordinary resources, important resources, competitive advantages, dynamic capacity, support competences, organizational competences, distinctive competences, and essential competences [36,41,42]. In a similar fashion, Bansal and DesJardine divided the triple-baseline into sets of indicators [43]. These sets entailed 19 economic, 23 environmental, and 32 social indicators. Furthermore, Van Marrewijk and Werre proposed an index ranging from 0 to 5 [44]. Level 0 indicates no regard for sustainability, Level 1 means meeting legal standards, Level 2 means following sustainability requirements for the sake of reputation, Level 3 involves balancing the triple-baseline, Level 4 shows innovation in sustainability, and Level 5 includes business strategies reflecting the interdependence of the environment, society, and economy.

### 2.3. Organizational Sustainability in Saudi Arabia

Few studies have assessed the extent of organizational sustainability in Saudi Arabia. Alharbi et al., for example, examined it with regard to total quality management (TQM) in Saudi hotels [45]. They found a direct correlation between TQM and overall organizational sustainability, where TQM was defined as striving to exceed customer expectations. In an updated study, researchers found that organizational climate significantly affected the organizational sustainability of hospitality organizations in Saudi Arabia, where the climate was described as the business atmosphere and work environment [46]. Soliman et al. claimed that organizational sustainability in Saudi Arabia could be increased with financial inclusion opportunities in small- to medium-sized enterprises [47]. In that study, financial inclusion entailed economic assistance programs for employees and consumers, such as financial planning, money management, and investment advising. Additionally, Kassem et al. found that transactional partnerships between charitable organizations played a role in intercompany sustainability frameworks, making organizational sustainability policies easier to draft [48].

In one study, business support significantly affected organizational sustainability while examining success factors in small- and medium-sized enterprises in Saudi Arabia [49]. Business support was defined as financial, governmental, and familial assistance granted to the company. Another study reported that organizational sustainability framework compliance in Saudi Arabia was highest when company members were rewarded or received social recognition for their sustainable measures [50]. Furthermore, Hashmi et al. proposed that Saudi organization leaders who attended newer universities often instilled more sustainability measures in their companies because education programs had begun teaching more sustainable business practices [51]. Vinodkumar and Ghadah reported that environmental social governance rates aligned with organizational sustainability, noting that female executives in Saudi Arabia showed higher environmental social governance and organizational sustainability in general [52]. Likewise, Saudi companies using knowledge-sharing processes in the form of knowledge donation and knowledge collection have reported higher organizational sustainability [53].

### 2.4. The Information System Success Model

The Information System Success Model (ISSM) measures the efficiency and effectiveness of an information system. DeLone and McLean proposed a six-dimensional information system model, also known as the DLML model, to address information system success in organizations [54]. The six dimensions consist of system quality, information quality, use, user satisfaction, individual impact, and organizational impact [55]. System quality here can be defined as the technical measures within the organization, such as

computer quality, response time, system accuracy, and ease of use [54,55]. Similarly, information quality encompasses organizational content with regard to relevance, timeliness, accessibility, adaptability, and accuracy. Use indicates connection time, number of computer functions used, and query time reported by system users, while user satisfaction is the impact of the information on the user. The individual impact is measured by how the user's experience was modified by the system, and organizational impact involves the overall influence of the system and its information on the organization [56,57]. System and information quality are independent foundations for the model, influencing the use and user satisfaction of the system. These four bases then create individual impact, which eventually influences the organization as a whole [54].

DeLone and McLean later proposed an updated model, adding the dimensions of service quality and net benefits (see Figure 1) [58]. The individual impact dimension was removed from the model, as multiple studies found that it strongly positively correlated with all other dimensions [59–61]. Intent to use was added to the previous dimension of use, as Seddon predicted its measurability to be more objective [62]. The aforementioned dimensions were redefined in the "10-year update" paper to provide better tools for future studies [58]. DeLone and McLean described system quality as desirable characteristics of an information system, such as system flexibility, reliability, intuitiveness, and sophistication [54]. Information quality similarly covers the conciseness, completion, and understandability of output events. The new dimension, service quality, indicates responsive and knowledgeable user support from IT personnel. Use and intention to use were described as how a system is utilized with respect to its operationalization, frequency, nature, extent, and purpose. Likewise, user satisfaction encompasses users' emotional response to system use. Lastly, net benefits are the additional success an information system brings on a larger scale, including individuals, groups, companies, work fields, or countries. Overall, studies utilizing the ISSM agree that the quality of features in an information system impacts the use of and user satisfaction with the system, ultimately determining the net benefits of that system [63–66]. The DLML model is a common method of measuring information system success [63–65].

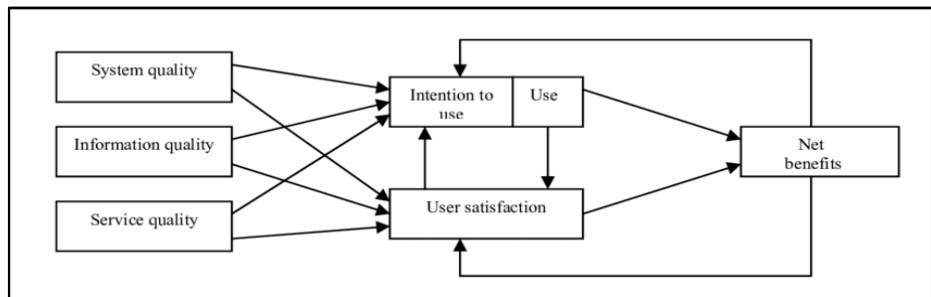

**Figure 1.** The Information Success Model [58].

Previous empirical research has found the ISSM to be efficient in measuring the success of an information system. Ojo, for example, found it was valid for studying the success of hospital information systems [67], while Hasselberg et al. confirmed the model's validity in health consultations through telemedicine [68]. Aldholay et al. confirmed the DLML model's effectiveness at measuring information system success in online education [69], while Wang et al. recognized the ISSM's effectiveness at analyzing information system success in technology-mediated learning [70]. Cui et al. used the ISSM to study international e-commerce, likewise validating the model [71]. Similarly, Ali et al. found the DLML model to be sufficient in measuring information system success with online retailers and transactions [72].

*2.5. Information System Success and Organizational Sustainability*

While there is a dearth of conceptual and empirical research linking information success to organizational sustainability, these constructs are related in several respects [28,58,73]. First, when information systems generate accurate, helpful, and rich data, stakeholders are more likely to make decisions that help the organization perform more effectively and efficiently. The ease of getting information and the usefulness of reports generated by information systems enable top management to recognize personnel problems. Reports, graphs, tables, and narratives created by such systems allow management to design working solutions, increasing the viability of the organization and economic sustainability.

System quality strengthens existing organizational sustainability practices [27,48,49,74]. On the one hand, having fewer technical problems enables employees to use the system to increase collaboration and communication at all levels. By the same token, system quality signals the credibility of an organization's commitment to its vision and mission, making employees more confident in their work. This arrangement motivates employees to actively engage in cultural and environmental sustainability practices at the individual level. The higher that information systems score on quality metrics, the more likely employees will use them, increasing the sharing of information across an organization, saving vital economic resources, and engaging the organization with community stakeholders.

Service quality helps achieve higher levels of organizational sustainability [63,67,75]. First, the reliability of services fosters trust among regular employees, customers, and top management. This increases the flow of helpful information, impacting effectiveness across all domains. Furthermore, the timeliness and efficacy of services increase employee commitment and loyalty to organizations, creating a more competitive environment. When employees observe the tangible benefits of information systems within their organization, they are more motivated to actively contribute, increasing social and cultural sustainability.

*2.6. Hypotheses Proposed in This Research*

- There is a positive, statistically significant relationship between net benefits and organizational sustainability.
- There is a positive, statistically significant relationship between user satisfaction and organizational sustainability.
- There is a positive, statistically significant relationship between system use and organizational sustainability.
- There is a positive, statistically significant relationship between service quality and organizational sustainability.
- There is a positive, statistically significant relationship between information quality and organizational sustainability.
- There is a positive, statistically significant relationship between system quality and organizational sustainability.

Figure 2 demonstrates the conceptual model to be tested in this study. Note that organizational sustainability is the main dependent variable. The information success model constitutes the main independent variables. These are system quality, information quality, service quality, system use, user satisfaction, and perceived net benefits. Additionally, several control variables were added to the model highlighted by prior research. These are personnel size, budget, diversity emphasis, information technology use, age of the agency, and female composition.

## 3. Methodology

*3.1. Research Design*

This cross-sectional correlational study employed quantitative questionnaires to determine the association between information system success and organizational sustainability. Quantitative data allowed the researchers to estimate the correlations between various dimensions measuring the two constructs. Self-reported data was ideal for assessing these dimensions across a sample of organizations. Given the lack of thorough lists of representa-

tive employees across either the public or private sector in Saudi Arabia, non-probability sampling was employed. Convenience sampling is a cost-effective approach to reach a large pool of participants. Given the limited resources, panel or longitudinal data collection would have been prohibitively difficult for this study. Instead, a cross-sectional design was employed to estimate the relationship between information system success and organizational sustainability at a single point.

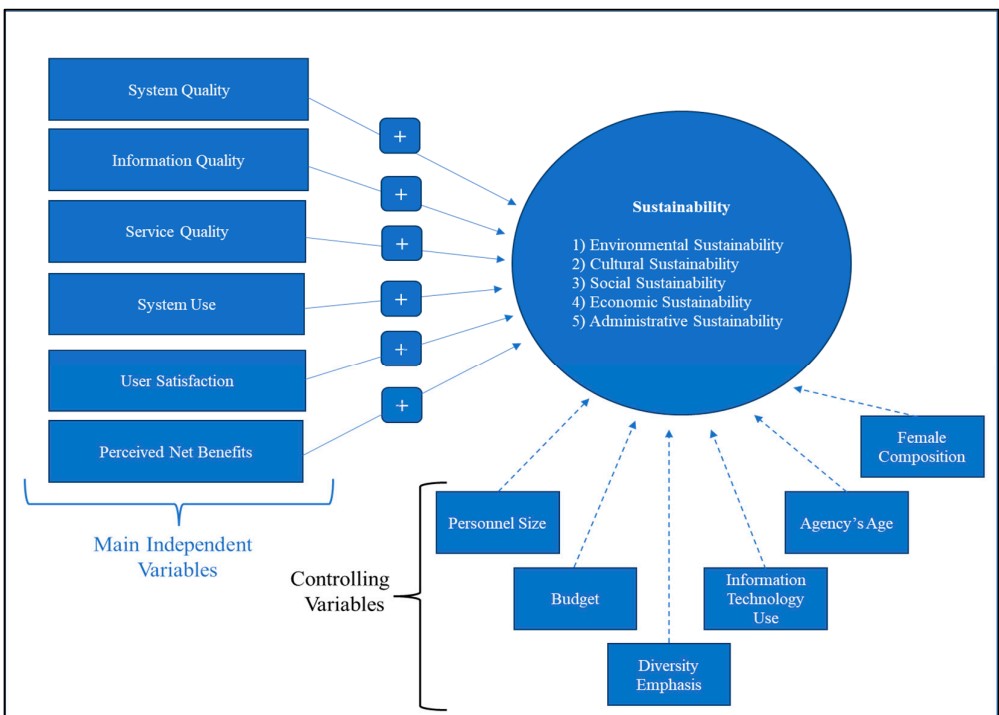

**Figure 2.** Conceptual model.

### 3.2. Population and Sampling

The population consisted of all active full-time salaried employees with Saudi citizenship working for the government of Saudi Arabia at the time of the study. There were 1,524,466 civil servants working for the Saudi government as of December 2021 [76]. This figure excludes military divisions and agencies. There were 561,671 employees on the general service schedule with varying grades and steps, 195,094 in the public health sector, 5636 in the justice system, 1390 diplomats in the Ministry of Foreign Affairs, 543,466 staff in K–12 education, 75,631 instructional staff in higher education, and 233,500 armed and civilian members of the military.

The researchers had no access to sampling frames containing all members in the intended population, as ministries and commissions were unwilling to provide thorough lists of employees due to legal, privacy, and security concerns. As a result, the study used non-probability sampling.

The researchers reached out to all public ministries and commissions, inviting them to participate in the project. More than 20 ministries and 50 public commissions were contacted to complete an online questionnaire. Ultimately, six ministries and seven commissions covering a wide range of industries and sectors agreed to participate. The sample consisted of 3738 employees representing a wide range of characteristics and many segments of the intended population. Table 1 summarizes the characteristics of the sample.

**Table 1.** Characteristics of the Sample.

| | |
|---|---|
| Age | Mean: 48.42<br>SD: 7.82<br>Min: 23<br>Max: 67 |
| Gender | Male: 67%<br>Female: 33% |
| Education | Less than an undergraduate degree: 8%<br>Undergraduate degree or equivalent: 39%<br>Graduate degree or higher: 53% |
| Experience | Less than 3 years: 21%<br>3 to 10 years: 36%<br>More than 10 years: 43% |
| Computer Proficiency | Minimal: 12%<br>Moderate: 31%<br>High: 57% |
| College Major | Information technology-related: 31%<br>Non-information technology-related: 69% |

### *3.3. Data Collection*

#### 3.3.1. Questionnaires

The independent variable was information system success, a multidimensional construct. Delone and McLean measured this construct using system quality, information quality, service quality, user satisfaction, perceived net benefits, and intended use [58]. This study employed a modified version of Tilahun and Fritz's 25-item questionnaire [77], validated in previous studies [75], to measure information system success. It measures system quality with four items covering employee perceptions of a system's ease of use, navigability, response time, and efficiency. Survey items are provided in the Appendix A Section of this study. It measures information quality using five items covering employee perceptions of the adequacy, availability, completeness, timeliness, and compatibility of information and reports generated by an information system. Service quality is measured using seven items capturing employee perceptions of information dependability, reliability of maintenance resources, responsiveness, and service consistency. System use is measured with two items: frequency of use and dependability to perform essential tasks at work. User satisfaction is measured with four items that quantify employees' perceptions of a system's ability to improve their work, facilitate their tasks, contribute to their productivity, and make them satisfied at work. Perceived net benefits are measured using three items covering employees' perceptions of the role of information systems in making their clients more satisfied, improving overall effectiveness/efficiency in their organization, and enhancing the quality of services provided by the organization. The instrument consists of 25 items asking employees to what extent they agree or disagree with a given statement. Each item is measured on a 4-point Likert scale: 0 (strongly disagree), 1 (disagree), 2 (agree), and 3 (strongly agree).

The dependent variable in this study was organizational sustainability, measured using a modified version of a 30-item questionnaire designed by Tilahun and Fritz to capture the multidimensional structure of the construct [77]. The first dimension, environmental sustainability, was measured using six items covering an organization's use of environmentally friendly materials, efficient use of resources, participation in community events to preserve the environment, and encouraging all employees to take action to live on a cleaner planet. The second dimension, cultural sustainability, was measured using five items covering employees' perceptions about their organization's evolving priorities, respect for diversity, commitment to inclusive representation, integration of all cultural elements, and active manifestation of everyone's culture through meaningful symbolism. Third,

social sustainability was measured using eight items regarding the integration of internal stakeholders into the organization and its commitment to external partners/customers. Statements measured the organization's effort to include everyone in the organization, train employees on being socially responsible toward each other and the outside environment, and partner with other organizations to advance social priorities like equal employment opportunity and equity. Fourth, economic sustainability was measured using four items regarding an organization's efficient use of its budget, resources, products, and materials. Fifth, administrative sustainability was measured using seven items covering employees' perceptions of the organization's innovative, flexible, and responsive structure to internal and external needs as well as its commitment to lean management, internal staff support, and integrity with customers. All items were measured on a 4-point Likert scale: 0 (strongly disagree), 1 (disagree), 2 (agree), and 3 (strongly agree).

The 30-item questionnaire also solicited employees' perceptions about vital characteristics of their organization, including personnel size, budget, age, commitment to diversity, and gender equity. All organization-level characteristics were controlling variables that could influence the pattern of associations linking information system success measures with organizational sustainability dimensions. The appendix provides the wording for each controlling variable and its values.

### 3.3.2. Distributing the Questionnaires

The researchers invited all government ministries to participate between November and December 2022. According to the most recent government tally, there were 24 active ministries in Saudi Arabia at the time of the study [78]. Invitations were sent to the chiefs of human resources in the Ministries of Defense; National Guard; Interior; Foreign Affairs; Communications and Information Technology; Municipal Rural Affairs; Economy and Planning; Industry and Mineral Resources; Energy; Investment; Commerce; Education; Culture; Social Development; Environment, Water, and Agriculture; Islamic Affairs; Pilgrimage; Sports; Health; Justice; Finance; Transportation; Media; and Tourism. Six ministries agreed to participate: the Ministries of Media; Culture; Communications and Information Technology; Commerce; Education; and Environment, Water, and Agriculture. The others cited legal, administrative, or privacy concerns and declined to participate. The human resources office staff distributed the questionnaires to internal lists inaccessible to the researchers. Two reminders were sent to staff, encouraging them to complete the questionnaires via a secure link to Qualtrics.

To increase representation, the researchers reached out to all active government commissions. As of the beginning of 2022, there were over 50 government commissions in Saudi Arabia. The researchers sent invitations to each commission's human resources office. Out of 50, seven agreed to participate without requiring the researchers to complete a large amount of paperwork. These commissions were the General Authority for Statistics, Saudi Geological Survey, Human Rights Commission, Digital Government Authority, Public Health Authority, Libraries Commission, and Education and Training Evaluation Commission. Human resources staff sent reminders to participate to encourage an increase in response rates.

The invitation email contained an information sheet in Arabic detailing the purpose of the study, anonymity of responses, and the use of responses for statistical purposes. Respondents were fully informed of their ability to cease participation without any consequences. They were also instructed to complete the questionnaires at their chosen place and time to guarantee their privacy. Given the online nature of the research, written signatures consenting to participate were not required by participating agencies or the researchers' institution. A local committee on research ethics from King Saud University approved this project.

*3.4. Data Analysis*

Reliability and Validity

This study employed two questionnaires—one for information system success and one for organizational sustainability—on a new population, Saudi public sector employees. Two sets of analyses were used to establish reliability. First, the internal consistency of each instrument was evaluated using Cronbach's alpha. The entire set of items was inserted into SPSS (Version 25) to generate reliability coefficients. Cronbach's alpha per instrument was assessed to evaluate the overall reliability of the instruments. If the value were 0.7 or higher, the instrument would be considered reliable for research purposes. Second, total correlations between each item on every instrument and the total scale were obtained to assess the stability of items. If an item's total correlation exceeded 0.3, the item would be considered stable.

In order to evaluate the validity of each instrument, exploratory factor analysis was conducted using SPSS. All items per instrument were included in each analysis. The rotated loadings matrix per analysis showing the correlations between each item and all extracted unobserved factors were retrieved from the output. Note that for an item to be valid, it must not demonstrate cross-loadings on more than a single factor. Each factor must exhibit a set of items with appropriate loading values exceeding 0.5. For the organizational sustainability questionnaire, five factors were expected to be observed: environmental, cultural, social, economic, and administrative dimensions. For the information system success questionnaire, six latent factors were expected: system quality, information quality, service quality, use, user satisfaction, and perceived net benefits.

In order to assess the effects of information system success on organizational sustainability, a series of multiple regression analysis models were fitted. A total of six models were estimated, each with the same set of independent variables and a separate dependent variable. The independent variables included individuals' demographic characteristics and their perceptions of the six information system success dimensions. The dependent variable in each model represented a different dimension of organizational sustainability. The sixth model featured the use of a summated scale for organizational sustainability, averaging scores on the five dimensions for participants.

Each model output reflected unstandardized coefficients, standard errors of the coefficients, and observed significance levels (*p*-values) along with diagnostic values, including variance inflation factors per coefficient. The *p*-value was used to determine the statistical significance of each coefficient. If the *p*-value on a coefficient was equal to or less than 0.05, the variable was statistically significant. From the regression output, effect sizes for each significant coefficient were calculated. This helps readers evaluate the practical significance of the results. An effect size of around 0.2 would indicate the variable had a small effect on the dependent variable, 0.5 would indicate it had a medium effect, and 0.8 would indicate a strong effect.

## 4. Results

Table 2 presents the results from the reliability and validity analysis for the information system success questionnaire. Overall, the instrument achieved acceptable reliability with a Cronbach's alpha of 0.85. By the same token, each item's total correlation was above 0.30, indicating the stability of all items. Note that the item total correlation represents the correlation between each item and the total score on the scale of all items without that item. Additionally, each set of items corresponding to a dimension in the instrument was included in a reliability analysis separate from all other sets of items. Consistent with the overall reliability of the instrument and the scores it generated, each construct had a Cronbach's alpha exceeding 0.70, indicating adequate reliability for research purposes.

**Table 2.** Reliability and Validity Estimates for Information System Success Instrument.

| Construct/Item | Item Total Correlation | Factor Loading |
| --- | --- | --- |
| **System Quality** ($\alpha$ = 0.78) | | |
| SyQ_01 | 0.52 | 0.61 |
| SyQ_02 | 0.47 | 0.50 |
| SyQ_03 | 0.34 | 0.56 |
| SyQ_04 | 0.63 | 0.72 |
| **Information Quality** ($\alpha$ = 0.81) | | |
| InQ_01 | 0.37 | 0.54 |
| InQ_02 | 0.47 | 0.64 |
| InQ_03 | 0.39 | 0.59 |
| InQ_04 | 0.57 | 0.74 |
| InQ_05 | 0.43 | 0.60 |
| **Service Quality** ($\alpha$ = 0.88) | | |
| SeQ_01 | 0.45 | 0.63 |
| SeQ_02 | 0.38 | 0.53 |
| SeQ_03 | 0.58 | 0.73 |
| SeQ_04 | 0.44 | 0.59 |
| SeQ_05 | 0.34 | 0.51 |
| SeQ_06 | 0.49 | 0.67 |
| SeQ_07 | 0.36 | 0.58 |
| **System Use** ($\alpha$ = 0.70) | | |
| SyU_01 | 0.52 | 0.81 |
| SyU_02 | 0.41 | 0.73 |
| **User Satisfaction** ($\alpha$ = 0.73) | | |
| UeS_01 | 0.57 | 0.78 |
| UeS_02 | 0.47 | 0.63 |
| UeS_03 | 0.50 | 0.67 |
| UeS_04 | 0.31 | 0.56 |
| **Perceived Net Benefits** ($\alpha$ = 0.76) | | |
| PNB_01 | 0.35 | 0.58 |
| PNB_02 | 0.46 | 0.63 |
| PNB_03 | 0.33 | 0.57 |
| Entire Instrument ($\alpha$ = 0.85) | | |
| N = 2973 | | |

Table 2 also shows the factor loadings of each item on its theorized unobserved latent factor. All items are loaded on their hypothesized dimensions. There were no cross-loadings exceeding 0.50 for any item in the rotated solution of the exploratory factor analysis. The initial solution generated six factors, each exhibiting a distinct nature with a specific set of items loading on it as theorized by the instrument. This suggests that the information system success instrument generated six dimensions as intended.

Table 3 displays the reliability and validity results of the organizational sustainability instrument. First, the instrument's overall reliability was assessed by including all items when calculating Cronbach's alpha. An internal consistency of 0.88 suggested sufficient reliability. Furthermore, each item's total item correlation was estimated to investigate its stability. All correlations exceeded 0.30, suggesting that each item was stable. Each set of items corresponding to their construct was included separately to calculate their Cronbach's alpha. Each construct's Cronbach's alpha exceeded 0.70, indicating the instrument and its data were reliable for the sample under consideration.

**Table 3.** Reliability and Validity Estimates for Organizational Sustainability Instrument.

| Construct/Item | Total Item Correlation | Factor Loading |
|---|---|---|
| **Environmental Sustainability** ($\alpha$ = 0.74) | | |
| EnS_01 | 0.33 | 0.54 |
| EnS_02 | 0.39 | 0.62 |
| EnS_03 | 0.34 | 0.55 |
| EnS_04 | 0.42 | 0.63 |
| EnS_05 | 0.32 | 0.51 |
| EnS_06 | 0.45 | 0.63 |
| **Cultural Sustainability** ($\alpha$ = 0.73) | | |
| CuS_01 | 0.36 | 0.56 |
| CuS _02 | 0.46 | 0.68 |
| CuS _03 | 0.43 | 0.63 |
| CuS _04 | 0.37 | 0.59 |
| CuS _05 | 0.34 | 0.53 |
| **Social Sustainability** ($\alpha$ = 0.79) | | |
| SOS_01 | 0.48 | 0.71 |
| SOS_02 | 0.41 | 0.67 |
| SOS_03 | 0.46 | 0.64 |
| SOS_04 | 0.40 | 0.611 |
| SOS_05 | 0.35 | 0.56 |
| SOS_06 | 0.31 | 0.50 |
| SOS_07 | 0.43 | 0.64 |
| SOS_08 | 0.31 | 0.58 |
| **Economic Sustainability** ($\alpha$ = 0.81) | | |
| ECS_01 | 0.53 | 0.75 |
| ECS_02 | 0.51 | 0.70 |
| ECS_03 | 0.48 | 0.67 |
| ECS_04 | 0.59 | 0.82 |
| **Administrative Sustainability** ($\alpha$ = 0.84) | | |
| ADS_01 | 0.55 | 0.78 |
| ADS_02 | 0.43 | 0.65 |
| ADS_03 | 0.59 | 0.82 |
| ADS_04 | 0.45 | 0.63 |
| ADS_05 | 0.52 | 0.74 |
| ADS_06 | 0.50 | 0.71 |
| ADS_07 | 0.42 | 0.59 |
| Entire Instrument ($\alpha$ = 0.88) | | |
| *N* = 2871 | | |

Table 3 presents factor loadings for each item to evaluate the validity of the instrument. Note that each item loaded on its theorized latent factor with a loading exceeding 0.50. None of the items loaded on another factor with a loading exceeding 0.50. The initial and rotated solutions generated a five-factor structure corresponding to the five dimensions of the instrument. These results again indicated the instrument and data were valid for the sample.

The exact value of acceptable factor loadings in Exploratory Factor Analysis is unclear. Some researchers [79–81] believe that if the sample size is large, a factor loading of 0.40 or higher is acceptable. Moreover, these researchers have argued that if cross-loadings are minimal, below 0.30, and factor loadings are 0.40 or larger, then the factor loadings are acceptable. Further, Stevens [79] suggested that irrespective of sample size, factor loadings of 0.40 or higher are acceptable for interpretive purposes. Therefore, in this research, all loadings are considered acceptable.

Figure 3 shows the amount of each type of sustainability reported by participants in Saudi public organizations. On a scale ranging from 0 to 100, with higher scores corresponding to more sustainability, self-reported responses indicated that Saudi institutions had moderate levels of overall organizational sustainability (*M* = 55.2) and administrative

sustainability (*M* = 55). Participants also reported reasonably high levels of cultural (*M* = 72) and social (*M* = 68) sustainability. In contrast, responses indicated low rates of economic and environmental sustainability.

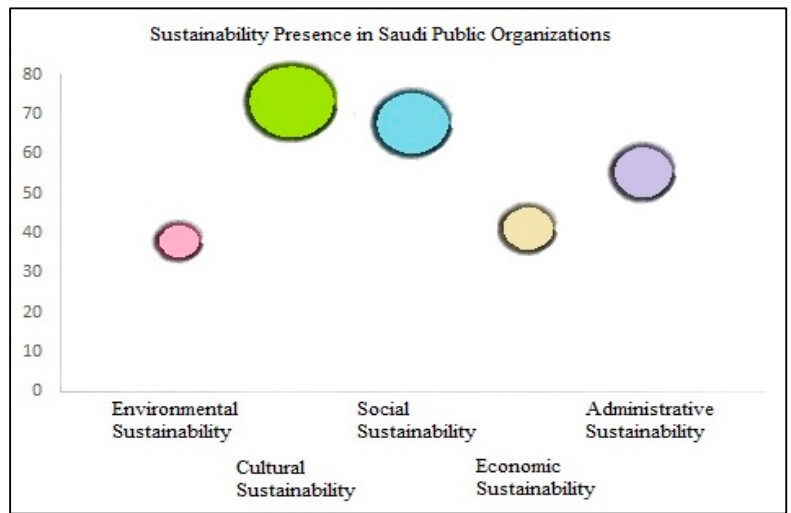

**Figure 3.** Organizational sustainability rates in Saudi public organizations.

Table 4 presents the bivariate correlations between types of organizational sustainability and the dimensions of information system success. Environmental sustainability was positively related to all information system dimensions. More importantly, it was significantly associated with user satisfaction and organizational net benefits. Cultural sustainability was not significantly correlated with any information success dimensions except user satisfaction. Social sustainability was only significantly positively related to service quality and user satisfaction, while economic sustainability was only significantly and positively related to user satisfaction. Noticeably, administrative sustainability showed a significant positive relationship to all three dimensions: service quality, user satisfaction, and net benefits.

**Table 4.** Correlations between Organizational Sustainability and Information Success Constructs.

| Variable | System Quality | Information Quality | Service Quality | System Use | User Satisfaction | Net Benefits | | |
|---|---|---|---|---|---|---|---|---|
| Environmental sustainability | 0.09 | 0.11 | 0.18 | 0.08 | 0.29 | 0.21 | | 1.00 / 0.80 |
| Cultural sustainability | −0.07 | 0.06 | −0.09 | −0.04 | 0.26 | 0.1 | | 0.60 / 0.40 |
| Social sustainability | 0.08 | 0.05 | 0.21 | −0.09 | 0.23 | 0.11 | | 0.20 / −0.20 |
| Economic sustainability | 0.09 | −0.12 | 0.12 | 0.14 | 0.22 | 0.13 | | −0.40 / −0.60 |
| Administrative sustainability | 0.06 | −0.11 | 0.26 | 0.17 | 0.28 | 0.41 | | −0.80 / −1.00 |

Note: All correlations are statistically significant at the 0.05 level.

Table 5 and Figure 4 present the results of the multiple regression analysis investigating the effects of information system success dimensions on organizational sustainability. Note that each dimension was transformed to a scale ranging from 0 to 100. For instance, environmental sustainability had six items ranging from 0 to 3. The summated scale resulting from averaging all items together ranged from 0 to 18. Then each score was multiplied by 5.556 to create the transformed scale ranging from 0 to 199. Similar transformations for each summated scale were conducted to standardize the interpretation of the analysis.

**Table 5.** Multiple Regression Analysis Predicting Organizational Sustainability Using Information System Success and Controlling Variables.

| Variable | Model 1 Environmental Sustainability | Model 2 Cultural Sustainability | Model 3 Social Sustainability | Model 4 Economic Sustainability | Model 5 Administrative Sustainability | Model 6 Overall Sustainability |
|---|---|---|---|---|---|---|
| System Quality | 0.94 (0.82) | −1.35 (2.83) | 1.02 (3.55) | 0.84 (1.23) | 1.07 (2.04) | 1.37 (1.02) |
| Information Quality | 1.47 (1.11) | 1.25 (2.98) | 0.94 (2.44) | −1.43 (3.24) | −1.84 (1.43) | −0.73 (1.32) |
| Service Quality | 2.12 * (0.91) | −0.64 (1.45) | 3.48 * (1.43) | 0.74 (0.94) | 0.63 (0.48) | 1.97 * (0.82) |
| System Use | 0.83 (1.64) | 1.03 (3.05) | −1.09 (2.33) | 1.34 (1.63) | 1.22 (0.86) | 1.12 (0.88) |
| User Satisfaction | 2.85 * (0.77) | 2.54 * (0.92) | 2.11 * (0.87) | 2.76 * (0.87) | 2.84 * (0.93) | 2.65 * (0.91) |
| Net Benefits | 1.94 * (0.81) | 1.93 (3.27) | 1.43 (2.25) | 0.42 (1.64) | 1.85 * (0.57) | 2.10 * (0.85) |
| Personnel Size | −1.84 * (0.79) | −3.01 * (1.23) | −3.07 * (1.23) | −2.49 * (0.95) | −2.39 * (0.98) | −2.26 * (0.73) |
| Budget | 2.17 * (0.90) | 3.53 * (0.94) | 2.38 * (1.09) | 2.97 * (1.02) | 2.11 * (0.82) | 2.86 * (0.90) |
| Diversity Emphasis | 1.84 * (0.68) | −2.34 (2.93) | 1.28 (2.32) | 1.83 (2.03) | 0.54 (1.03) | 1.76 * (0.61) |
| Information Technology Use | −0.76 (1.43) | −2.35 (3.21) | −2.23 (3.21) | 0.94 (0.69) | 1.31 (1.87) | 0.45 (0.87) |
| Agency's Age | 1.21 (1.88) | −2.43 * (0.84) | 1.65 (2.85) | −0.73 (1.02) | 0.84 (0.76) | −1.87 * (0.73) |
| Female Composition | −0.98 (1.53) | 1.09 (0.85) | 2.21 (2.66) | 0.58 (0.43) | 0.47 (0.38) | 0.68 (0.64) |
| R-Squared | 0.09 | 0.06 | 0.08 | 0.04 | 0.05 | 0.12 |
| N | 2093 | 2325 | 2125 | 2317 | 2285 | 1894 |
| VIFs | <2 | <2 | <2 | <2 | <2 | <2 |

Note: Statistical significance level is denoted by * at 0.05.

Model 1 features environmental sustainability as the dependent variable. The model explained about 9% of the variance within environmental sustainability, as indicated by the R-squared. System quality, information quality, and system use were not statistically significant predictors of employees' perceptions of their organization's environmental sustainability practices. On the other hand, service quality, user satisfaction, and net benefits were significant at the 0.05 level. Organization size with respect to personnel, operating budget, and emphasis on diversity were also significant predictors. All variance inflation factors for estimated coefficients were less than two, presenting no threat to multicollinearity in the model.

To interpret Model 1, it is important to keep in mind that information system success dimensions and environmental sustainability were measured on scales ranging from 0 to 100. For every unit increase in service quality, there was a 2.12-unit increase in environmental sustainability; for every unit increase in user satisfaction, there was a 2.85-unit increase in environmental sustainability; and for every unit increase in net benefits, there was a 1.94-unit increase in environmental sustainability. Organizations with more personnel experienced less environmental sustainability. In contrast, those with larger budgets and those that emphasized diversity were more likely to reportedly possess higher levels of environmental sustainability. Direct effects should be interpreted as holding other variables constant.

Model 2 features cultural sustainability as the dependent variable. Overall, the model explained 6% of the variance within Saudi public employees' perceptions of cultural sustainability. User satisfaction was the only information system success dimension significant

at the 0.05 level in predicting cultural sustainability. Personnel size, budget, and age were significant predictors as well.

For every unit increase in user satisfaction, there was a 2.54-unit increase in cultural sustainability. Larger organizations had lower levels of perceived cultural sustainability, while newer organizations and those with larger budgets tended to show higher levels of perceived cultural sustainability. Direct effects should be interpreted as holding other variables constant.

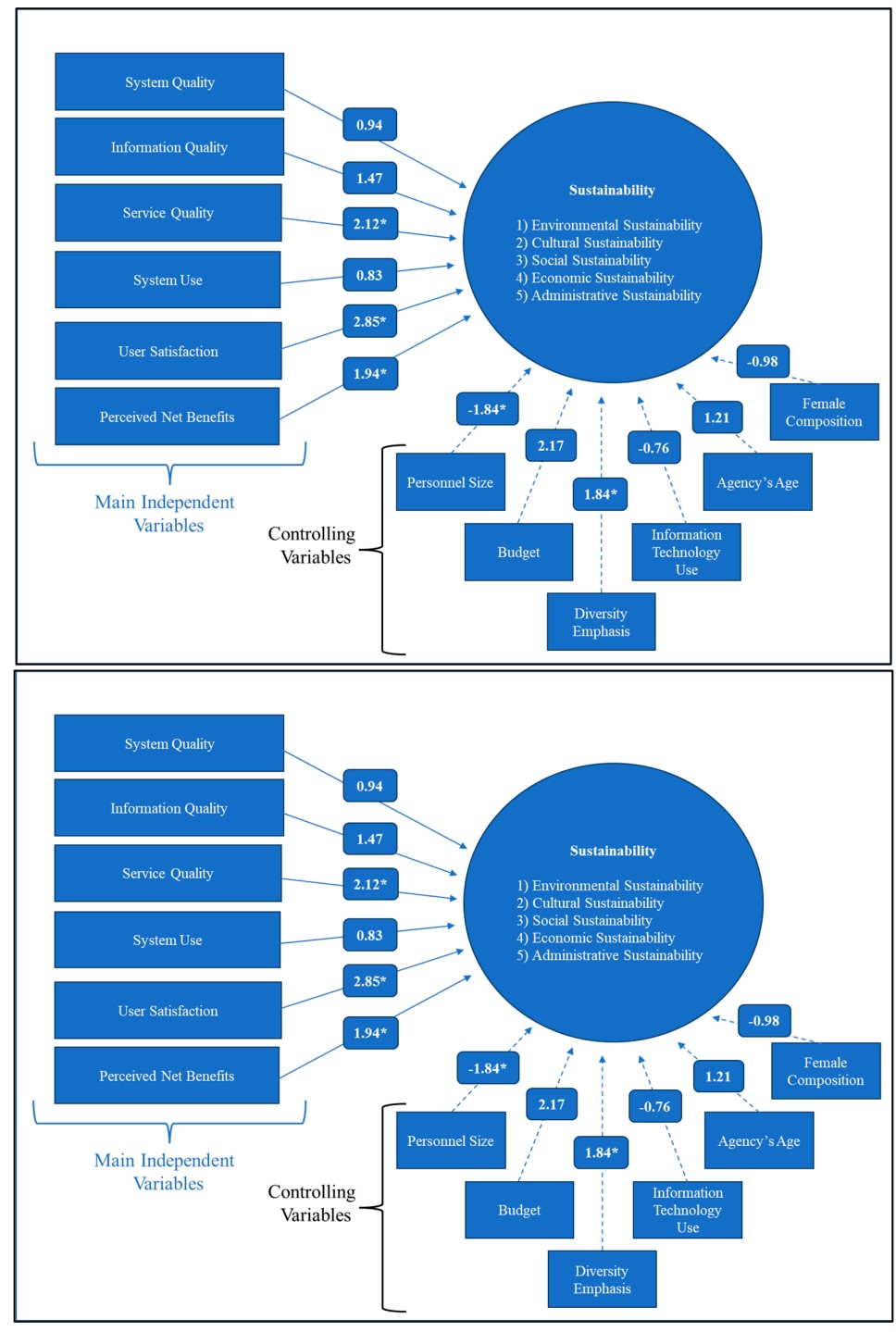

**Figure 4.** Conceptual model results (note that statistical significance level is denoted by * at 0.05.).

Model 3 features social sustainability as the dependent variable. Overall, the model explained about 8% of the variance in Saudi public employees' perceptions of their or-

ganization's social sustainability. Service quality, user satisfaction, personnel size, and budget size were significant predictors of perceived social sustainability. System quality, information quality, system use, and net benefits were not significant predictors.

For each additional unit in service quality, there was a 3.58-unit increase in perceived social sustainability, and for each unit increase in user satisfaction, there was a 2.11-unit increase in perceived cultural sustainability, holding other variables constant. Organizations with more personnel showed lower perceived social sustainability, while those with more financial resources were more likely to be seen as possessing higher social sustainability.

Model 4 features economic sustainability as the dependent variable. Overall, the model explained 4% of the variance in Saudi public employees' perceptions of economic sustainability. System quality, information quality, service quality, system use, net benefits, organization age, gender composition, diversity emphasis, and information technology were not significant predictors of perceived economic sustainability. User satisfaction, personnel size, and budget were significant predictors.

For each unit increase in user satisfaction, reported economic sustainability increased by 2.76; for each unit increase in personnel size, economic sustainability decreased by about 2.5. For each unit increase in budget, there was about a three-unit increase in perceived economic sustainability, holding other variables constant. All variance inflation factors on coefficients were less than two, indicating the absence of multicollinearity as a threat to the model's estimates.

Model 5 features administrative sustainability as the dependent variable. Overall, the model explained 5% of the variance in administrative sustainability perceptions. User satisfaction, net benefits, personnel size, and budget size were significant predictors of perceived administrative sustainability. System quality, information quality, service quality, system use, organization age, the proportion of female employees, diversity emphasis, and information technology use were not significant predictors.

For each unit increase in user satisfaction, perceived administrative sustainability increased by 2.84; for each unit increase in net benefits, administrative sustainability increased by 1.85; for each unit increase in personnel size, administrative sustainability decreased by 2.39; and for each unit increase in budget, administrative sustainability increased by 2.11, holding other variables constant. All variance inflation factors were less than two, indicating a minimal threat of multicollinearity to the estimates of the model.

Model 6 features an overall organizational sustainability scale comprising all five dimensions as the dependent variable. Overall, the model explained 11% of the variance in Saudi public employees' perceptions of organizational sustainability. System quality, information quality, system use, the proportion of female employees, and information technology were not significant predictors. User satisfaction, service quality, net benefits, personnel size, budget size, emphasis on diversity, and organization age were significant.

For each unit increase in service quality, perceived organizational sustainability increased by 1.97; each unit increase in user satisfaction was associated with a 2.65 increase in organizational sustainability; and each unit increase in net benefits reflected a 2.10 increase in organizational sustainability, holding other variables constant. Organizations with more personnel were less likely to be perceived to have organizational sustainability practices. In contrast, those with larger budgets, those that emphasized diversity, and newer organizations were more likely to be perceived to practice organizational sustainability.

Figure 5 displays the effect sizes per significant variable within each fitted model. Note that none of the variables had a strong effect on any type of sustainability, defined as a Cohen's *d* of 0.8 or greater. A few variables had a medium effect on sustainability, defined as a Cohen's *d* of 0.5–0.8. User satisfaction had a medium effect on environmental sustainability, personnel size had a moderate effect on cultural sustainability, service quality had a medium effect on social sustainability, and budget had a moderate effect on administrative sustainability. All other effects across all models featured small effects, defined as a Cohen's *d* of less than 0.5.

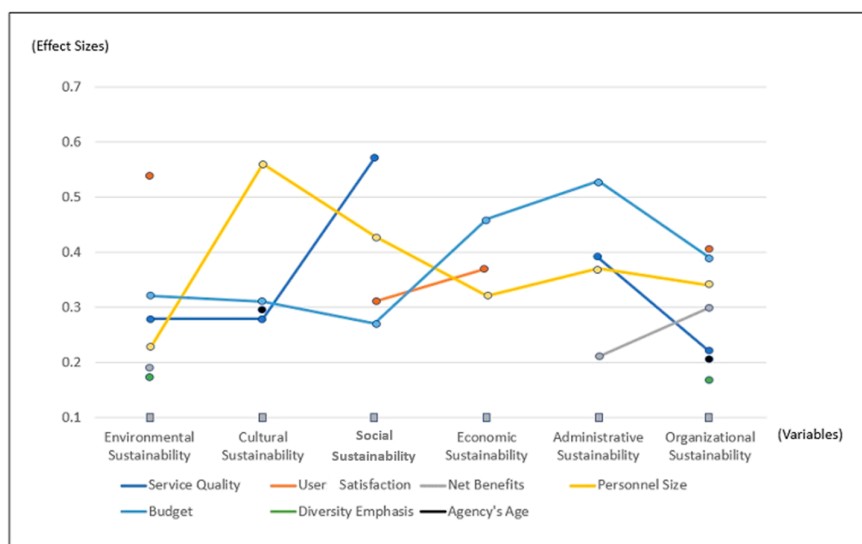

**Figure 5.** Variables' effect sizes.

## 5. Discussion

### 5.1. Relevance to Past Research

The findings in this research established direct associations between organizational characteristics and organizational sustainability practices. Similar to Batista and Francisco, who reported that organizations with more resources tended to invest in sustainability practices [5], this study found that organizations with larger budgets tended to be perceived as having a greater commitment to environmental, cultural, social, economic, and administrative sustainability practices. On the other hand, Batista and Francisco reported that older organizations were more mature and therefore generated more detailed sustainability reports compared to newer organizations [5], a finding that contradicted this study. Participants were more likely to perceive newer Saudi public organizations as implementing sustainability practices. One potential explanation for this contradiction is that the government of Saudi Arabia has hired thousands of new graduates with Western degrees in the past decade in newly established departments and agencies. These individuals have been exposed to more sustainability education and practices than their older peers, who attended college from the 1970s to the early 2000s in domestic institutions.

The findings revealed a weak association between technology implementation and organizational sustainability practices. More specifically, employees' perceptions of information system success provided little explanatory power in predicting their employers' organizational sustainability practices. User satisfaction with information systems was the most significant predictor of perceived organizational sustainability overall. A potential explanation for this is that when employees are satisfied with their job, including the systems they use every day, they are more likely to participate in organizational events and comply with recommended best practices that intersect with organizational sustainability. Notably, organizations with more employees reportedly had less organizational sustainability regardless of the type and age of the organization.

One of the most consistent findings throughout this study was that user satisfaction was related to all dimensions of organizational sustainability. While the study was concerned with user satisfaction with information systems, this satisfaction is part of overall employee satisfaction. Management and organizational research have established a robust direct relationship linking satisfaction with desirable outcomes like organizational commitment, loyalty, and citizenship [82,83]. Sustainability practices featured in the questionnaires asked participants if they were involved in all types of sustainability events. If employees were satisfied with their work, they were more willing to contribute to improving the quality of their organization. Psychological contract theory postulates that if employees believe their employers are serious about creating conducive work conditions for them

to succeed, they become more engaged in all activities across the organization [84]. As a result, stakeholders could introduce satisfaction initiatives in public organizations in Saudi Arabia to improve organizational sustainability.

Past research has neglected the relationship between technology implementation and organizational sustainability. Generally, organizations with richer technological infrastructures feature more sustainable practices compared to others. For instance, information technology businesses like Apple, Facebook, or Amazon are leaner than construction or logistics companies. The current findings suggest that newly established institutions tend to be more technologically oriented and, therefore, practice greater sustainability compared to all agencies.

### 5.2. Theoretical Implications

This study broadens the theoretical underpinnings of organizational sustainability to encompass new areas, such as cultural and administrative organizational practices. This is in line with prior calls for redefining the concept. For instance, Zawawi and Abd Wahab argued that corporate spirituality—defined as preparing decision-makers to become wiser in their choices in fostering lean management while preserving employee morale—should be a theoretically distinct factor in organizational sustainability [85]. Similarly, Horak et al. advocated for a more culturally relativistic and flexible conceptualization [86]. The current study presented the reliability and validity metrics for a culturally sensitive instrument measuring organizational sustainability beyond the classical triple-bottom-line view of the concept. The researchers altered the wording of items to make them suitable to the local context, which was supported by the appropriate validation of the instrument in Saudi Arabia, a previously neglected context in theorizing about organizational sustainability.

The present analysis departs from the institutional theoretical perspective on organizational sustainability. Instead of limiting organizational sustainability to cost reduction and profit maximization measures, it extended the construct to cover dimensions unrelated to the organization-level view of sustainability [87]. Furthermore, it included the human element in the understanding of sustainability in organizations, contrary to economic views of the concept that focus more on the structural and functional dimensions of organizational practices [88].

### 5.3. Practical Implications

Managers, supervisors, and division chiefs could use the findings of this study to improve organizational sustainability practices in their institutions in many ways. Batista and Francisco recommended paying attention to specific dynamics in a workplace when designing and implementing organizational sustainability frameworks [5]. In this research, direct positive associations were established between service quality, user satisfaction, and net benefits with overall organizational sustainability. Based on this set of associations, stakeholders could maximize service quality, perceived user satisfaction, and net benefits associated with any existing or future information system to enhance organizational sustainability practices. Relatedly, Smith argued that better understanding among people in organizations created work environments that fostered satisfaction, which as this research established, supports organizational sustainability [89].

A consistent finding across models in this research was that organizations with fewer employees showed more perceived organizational sustainability. Stakeholders in Saudi Arabia are therefore urged to restructure the division of labor and specification of functions within their public institutions. Prior analyses have pointed to bureaucracy preventing Saudi government agencies from implementing more timely changes [90]. In contrast, smaller organizations with a younger workforce have appeared more conducive to organizational sustainability in all its forms [91].

### 5.4. Future Research Directions

One promising research area in organizational sustainability is the connection between information technology and sustainability outcomes. Notwithstanding this study's mixed results on the positive influence of information success on employees' perceptions of different types of sustainability, future researchers may test various information system variables using different measures of sustainability. For instance, industry 4.0 technology deployment like machine learning, data analytics, cloud computing, or robotics could foster more sustainable practices at the organizational level [92–101]. In addition, game theory approaches can offer models to understand how interactions between actors influence outcomes [102–104]. In particular, modeling organizational actors to understand the connection between information technology and organizational sustainability could be a significant contribution to the literature. It is also important to note that exposure to information technology training may result in the adoption of sustainable practices at the individual employee level. The area where information technology and sustainability intersects shows growing potential for research and development.

The links between information technology variables and organizational sustainability practices need to be more fully explored by researchers and practitioners. Theoretical explanations behind user satisfaction, net benefits, and organizational sustainability dimensions are still emerging. Conjectures on how information technology may foster sustainability practices in organizations need to be validated using multi-methods and modeling techniques. Moderation and mediation analyses could help in discerning how information system variables explain changes in sustainability practices.

### 5.5. Limitations

The conceptualization and measurement of organizational sustainability or information system success may influence observed empirical results. In this study, organizational sustainability was measured using five dimensions: environmental, cultural, social, economic, and administrative sustainability. In a review of the literature, Nawaz and Koç conceptualized sustainability as a nine-factor construct: minimization of waste, operational excellence, corporate citizenship, research and innovation, logistics, governance, sustainability management, employee relations, and health and wellness. By the same token, the current study measured information system success using a six-dimensional structure: system quality, information quality, service quality, user satisfaction, intended use, and net benefits [40]. Alternatively, Gable et al. measured it with four dimensions: individual impact, organizational impact, system quality, and information quality [105]. The choice of models, items, data collection instruments, and dimensional structures directly affects observed associations between information system success and organizational sustainability.

The present study has several limitations. First, convenience sampling does not guarantee the generation of representative samples. Second, the regression models included certain sets of relevant variables. Earlier studies cited other variables that could influence organizational sustainability, including institutional industry, type of workforce, previous exposure to sustainability, information technology, data-driven innovation, and overall education level of employees [5,85,106–112]. Failing to include some variables introduces a source of error in the estimated coefficients. Third, cross-sectional designs are not appropriate for generating causal statements about the associations between variables in the analysis [113,114].

### 5.6. Conclusions

The study found only a weak link between corporate sustainability strategies and their use of technology. More specifically, employers' organizational sustainability policies were poorly predictable from employee ratings of information system success. User satisfaction with information systems was the most important indicator of sustainability across the organization. This may be explained by the fact that happy employees are more likely to participate in corporate activities and follow organizational sustainability-related best practices. This is true

both at work and in the systems we use every day. In particular, regardless of company type or age, companies with more employees were found to be less sustainable.

Organizational sustainability, therefore, provides companies with a range of social, economic, and environmental benefits. The study found that European companies' improved sustainability use improved their financial and market performance by 43.2%. When it comes to stakeholder and customer satisfaction, companies with social responsibility programs performed four times better than those without obvious community ties. Organizations that disclosed their sustainability practices saw a 55% increase in employee morale, a 43% increase in business process efficiency, and a 38% increase in employee loyalty. Another study found that companies were more successful in introducing tax cuts and subsidies to ensure environmental sustainability. Additionally, reducing greenhouse gas emissions can help ensure the viability of your business for future generations.

**Author Contributions:** Conceptualization, A.A. (Abdullah Almuqrin), I.M., J.Z.Z. and A.A. (Abdulaziz Alomran); methodology, A.A. (Abdullah Almuqrin), J.Z.Z. and I.M.; formal analysis, A.A. (Abdullah Almuqrin), I.M. and J.Z.Z.; writing—original draft preparation, A.A. (Abdullah Almuqrin), J.Z.Z. and A.A. (Abdulaziz Alomran); writing—review and editing, I.M., A.A. (Abdullah Almuqrin) and J.Z.Z. All authors have read and agreed to the published version of the manuscript.

**Funding:** This research was funded by the Researchers Supporting Project at King Saud University.

**Institutional Review Board Statement:** The study was conducted in accordance with the Declaration of Helsinki and approved by the Institutional Review Board (Human and Social Researches) of King Saud University (Ref No: KSU-HE-18-242).

**Informed Consent Statement:** Informed consent was obtained from all participants involved in this study. Therefore, an online agreement that all participants agree to by checking a box that says "I agree" before starting the questionnaire.

**Data Availability Statement:** Not applicable.

**Acknowledgments:** This research was funded by the Researchers Supporting Project number (RSP2023R453), King Saud University, Riyadh, Saudi Arabia.

**Conflicts of Interest:** The authors declare no conflict of interest.

## Appendix A

**Table A1.** Questionnaires.

| Variable | Item |
| --- | --- |
| System Quality | Information Systems at my agency are easy to use |
| | Information Systems at my agency is user friendly |
| | I find it easy to get Information Systems at my agency to do what I want them to do |
| | The response time of Information Systems at my agency is acceptable |
| Information Quality | Information Systems at my agency provide sufficient information to enable you to do your tasks |
| | I am satisfied with the accuracy of Information Systems at my agency |
| | Information Systems at my agency generate a complete report |
| | With Information Systems at my agency, I can access the information I need on time |
| | The reports from other departments are in the format of my need. |

**Table A1.** *Cont.*

| Variable | Item |
| --- | --- |
| Service Quality | Information Systems at my agency are dependable |
| | My supervisor has been helpful in the use of Information Systems at my agency |
| | The available user guides and help function is helpful |
| | User support for Information Systems at my agency is timely |
| | I can access computers in the ward when I need them |
| | The Information Systems at my agency automatically save records |
| | The reported bugs on the software get fixed in an acceptable time frame |
| Use | I frequently use Information Systems at my agency for my tasks |
| | I am dependent on Information Systems for my job tasks |
| User Satisfaction | I can finish my task faster with Information Systems |
| | Information Systems at my agency improve my productivity |
| | Information Systems at my agency have had positive impacts on the quality of my task |
| | Overall, I am satisfied with Information Systems at my agency |
| Perceived Net Benefit | I expect Information Systems at my agency to make client care faster |
| | I expect Information Systems at my agency to increase my effectiveness |
| | I expect Information Systems at my agency to make the agency services better |
| Computer Literacy | I am interested in working with computers |
| | I have moderate skills in using computers |
| | I take computer training in the hospital |
| | I am playful with technology |
| | I feel that using computers will support me to be more efficient in the future |

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
