# Peer review of "Information System Success for Organizational Sustainability: Exploring the Public Institutions in Saudi Arabia"

_sustainability, doi:10.3390/su15129233_

Round 1

Reviewer 1 Report

Appreciation to the authors for such a great work. However, I noticed few things which need more clarification. Concers are as follows:

1. Add Research Questions/Objectives at the end of the Introduction Section.

2. Second last paragraph in the introduction section seems to be aprt of discussion section. Add this paragraph in the introducition. (Line 85-94)

3. Authros have conduted literature review in detail; however it would eb betetr if authros add hypotheses at the end of the literature review

4. Please mention in methods section which ethics/research committee has approved this study in Saudi Arabia

5. Add Average vriance extracted and Compositie reliability in Table 3

6. Som Item total correlation are less than 0.4 however if overall Alpha is high than 0.7 it is up to authors to retain the item or delete it.

7. in Table 4 use * or ** or *** for significant relationships so that readers may find it easy to understand it.

8. In discussions ection compare your findings with past studies in order to justify your hypotheses.

English has no issues, however author can proof read it before publication.

Author Response

Reviewer 1

Comments

Replies

1. Add Research Questions/Objectives at the end of the Introduction Section.

Done. Please see the end of the introduction section.

2. Second last paragraph in the introduction section seems to be a aprt of discussion section. Add this paragraph in the introduction. (Line 85-94)

Done. The paragraph was moved, as you suggested.

3. Authors have conducted literature review in detail; however, it would eb better if authors add hypotheses at the end of the literature review

Done. A new section for hypotheses was added after the literature review section.

4. Please mention in methods section which ethics/research committee has approved this study in Saudi Arabia

Done. Please see the end of the data collection section.

5. Add Average variance extracted and Composite reliability in Table 3

This study uses exploratory factor analysis since the measurement tools have never been applied to the setting of Saudi Arabia. Further, the alpha coefficients presented led to the conclusion that the instrument is reliable for research purposes. Similarly, the factor loadings provided suggest that the instrument is valid for research purposes. Therefore, adding average variance extracted (AVE) and composite reliability (CR) is inappropriate because those values are associated with confirmatory factor analysis.

6. Some Item total correlation are less than 0.4 however if overall Alpha is high than 0.7 it is up to authors to retain the item or delete it.

Any item total correlation of .3 or higher suggests item stability, according to Andy Field’s book titled Discovering Statistics Using IBM SPSS Statistics.

7. in Table 4 use * or ** or *** for significant relationships so that readers may find it easy to understand it.

Done. Please see the note provided at the end of Table 4.

8. In discussions section compare your findings with past studies in order to justify your hypotheses.

Done. Please see the added part of the discussion section.

Reviewer 2 Report

The authors provide good background on the organizational sustainability and information system success in private organizations but not public organizations. Since the context of the paper is public institutions, is there any previous studies on public institutions. If there is, it is recommended that the authors added some studies on this.

Justify why convenience sampling is used as the sampling method.

Suggest to add the Conclusion section after Limitations.

It is recommended that the authors to remove or relocate the second last paragraph of the Introduction section i.e line 84-95. This paragraph describes the findings of the research which even though they were already covered in the Results section.

Author Response

Reviewer 2

Comments

Replies

1.   The authors provide good background on the organizational sustainability and information system success in private organizations but not public organizations. Since the context of the paper is public institutions, is there any previous studies on public institutions. If there is, it is recommended that the authors added some studies on this.

Research findings on the private sector are similar to those on the public sector. Organizational sustainability prevalence has not been linked to sectoral differences other than those included in the literature review section.

2.   Justify why convenience sampling is used as the sampling method.

The justification is included in the research design section, as seen below:

Given the lack of thorough lists of representative employees across the public and private sectors in Saudi Arabia, non-probability sampling was employed. Convenience sampling is a cost-effective approach to reach a large pool of participants. Given the limited resources, panel or longitudinal data collection would have been prohibitively difficult for this study. Instead, a cross-sectional design was employed to estimate the relationship between information system success and organizational sustainability at a single point.

3.   Suggest to add the Conclusion section after Limitations.

Done. The conclusion section was created, as you pointed out.

4.   It is recommended that the authors to remove or relocate the second last paragraph of the Introduction section i.e line 84-95. This paragraph describes the findings of the research which even though they were already covered in the Results section.

Done. The second last paragraph of the introduction section was removed and added to the discussion section.

Reviewer 3 Report

I feel happy to review this manuscript, titled as "Information System Success for Organizational Sustainability: 2 Exploring the Public Institutions in Saudi Arabia." The overall quality of this manuscript is fine. Although there are some changes are required in this manuscript. I hope authors will address all changes in the revised manuscript.

1. The introduction section should be revised, lines 85 to 94 explained about the findings of this study, no need to discuss findings in the introduction section. Also I suggest authors to clearly discuss the research gap with prior researchers studies on this topic. 

2. I suggest authors to add research questions in the introduction section.

3. There is no hypotheses relationships are proposed in this manuscript? 

4. I suggest authors to add sample items in the measure section, I am confused that may this data having issue of common method bias, because authors have adopted many survey items for data collection, How authors have managed this issue? Some of the measurement constructs factor loading is less than 0.70, is it acceptable? Does authors have performed EFA analysis or CFA analysis?

5. How the authors have tested the constructs validity ? Which test authors have done convergent or divergent validity ? I want to see the AVE values for each variables.

6. Why each model explained very less value of variance? 

7. There is so misunderstanding in Figure 1, authors have write two times "use". 

Good Luck

Author Response

Reviewer 3

Comments

Replies

1. The introduction section should be revised, lines 85 to 94 explained about the findings of this study, no need to discuss findings in the introduction section. Also, I suggest authors to clearly discuss the research gap with prior researchers studies on this topic.

Done. The second last paragraph of the introduction section was removed and added to the discussion section.

2. I suggest authors to add research questions in the introduction section.

Done. Please see the end of the introduction section.

3. There is no hypotheses relationships are proposed in this manuscript?

Done. A new section for hypotheses was added after the literature review section.

4. I suggest authors to add sample items in the measure section, I am confused that may this data having issue of common method bias, because authors have adopted many survey items for data collection, how authors have managed this issue? Some of the measurement constructs factor loading is less than 0.70, is it acceptable? Does authors have performed EFA analysis or CFA analysis?

Done. Survey items were added to the appendix. Some items were reverse coded when presented to participants to mitigate common methods bias problems. Yes, factor loadings are acceptable if they are 0.4 or above, as mentioned by Sawilowsky’s article titled New effect size rules of thumb.

5. How the authors have tested the constructs validity? Which test authors have done convergent or divergent validity? I want to see the AVE values for each variable.

Adding average variance extracted (AVE) is inappropriate because those values are associated with confirmatory factor analysis. This study uses exploratory factor analysis since the measurement tools have never been applied to the setting of Saudi Arabia.

Further, the alpha coefficients presented led to the conclusion that the instrument is reliable for research purposes. Similarly, the factor loadings provided suggest that the instrument is valid for research purposes.

Concerning the type of construct validity performed, exploratory factor analysis output could be used to evaluate divergent and convergent types of validity. Both types were satisfied based on the interpretations provided in the results section.

6. There is so misunderstanding in Figure 1, authors have write two times "use".

“Intention to use” is different from the term “use” in Figure 1. Use refers to actual use. The diagram was reproduced from DeLone and McLean’s study.

DeLone, W.H.; McLean, E.R. The DeLone and McLean model of information systems success: A ten-year update. J Manag Inf Syst 2003, 19(4), 9–30. doi:10.1080/07421222.2003.11045748.

Round 2

Reviewer 3 Report

Dear authors, I have seen many improvements in the revised manuscript. Figure 1 and the hypotheses which were stated are not aligned with the model. I suggest authors to add a conceptual model Figure, so I will understand the theme of this research and make some decision. Moreover, I have seen the factor loading values are less than 0.70, is there any issue with the data. Does authors perform EFA analysis or check the common method bias in the data?   

Minor editing of English language required.

Author Response

please see the attachment, thank you.

Round 3

Reviewer 3 Report

Authors have addressed all the comments in the revised manuscript.